# Table Tennis in Physical Education: Teachers’ Perceptions of Health-Related Aspects in School-Age Children

**DOI:** 10.3390/bs15111495

**Published:** 2025-11-04

**Authors:** Miguel Ángel Ortega-Zayas, Pamela Patanè, Carlos Peñarrubia-Lozano, Francisco Pradas

**Affiliations:** 1ENFYRED Research Group, University of Zaragoza, 22001 Huesca, Spain; maortega@unizar.es (M.Á.O.-Z.); carlospl@unizar.es (C.P.-L.); 2Motor Sciences, Department of Public Health, Experimental and Forensic Medicine, University of Pavia, 27100 Pavia, Italy; pamela.patane95@gmail.com

**Keywords:** racket sports, physical education, primary schools, teacher perceptions, health promotion

## Abstract

Table tennis (TT) is recognized for its accessibility, adaptability, and health benefits, making it suitable for physical education (PE). This study aimed to investigate the prevalence of TT implementation in primary school PE classes and explored associations with teacher characteristics and perceptions regarding injury risk, safety, inclusivity for students with disabilities or special educational needs, student engagement, and the educational value in PE curricula. A cross-sectional survey was conducted among 393 Spanish primary school PE teachers who completed the validated Racket Sports Attitude Scale (RSAS). Associations between teacher characteristics and TT use were tested using Pearson’s Chi-square, with effect sizes reported as Cramer’s V, Phi, and Somers’ D for ordinal variables. Additional analyses examined links between TT adoption and perceptions of injury risk, inclusivity, safety for pupils, ability to enhance engagement, and its educational value in PE. Only 11.7% of teachers reported using TT in PE classes. Implementation rates were not significantly different by sex but were associated with age (χ^2^ = 27.2, *p* < 0.001, Somers’ D = 0.071) and teaching experience (χ^2^ = 30.0, *p* < 0.001, Somers’ D = 0.099). TT use showed strong associations with perceptions of lower injury risk (Cramer’s V = 0.707), suitability for students with disabilities (0.712), special educational needs (0.715), safety (0.707), engagement (0.712), and educational value (0.716) (*p* < 0.001). Despite positive perceptions, TT is underutilized in PE curricula.

## 1. Introduction

Physical education (PE) represents a cornerstone of comprehensive child and adolescent development, providing benefits that extend beyond physical health to encompass cognitive, social, emotional, and motor domains ([16]; [23]; [35]; [46]). Well-structured PE programs foster lifelong habits of physical activity, promote social integration, and contribute to the prevention of chronic diseases ([8]; [13]; [31]). Within this context, the methodology with which they are presented, with a special focus on play as an introduction to sports, is decisive, as is the selection of sports disciplines in PE curricula is a critical pedagogical decision that can shape students’ engagement, skill acquisition, and attitudes toward physical activity ([9]; [12]; [33]; [43]).

Racket sports, including tennis, padel, badminton, and table tennis (TT), have been increasingly recognized as valuable educational tools due to their accessibility, adaptability, and ability to enhance both fundamental and sport-specific motor skills ([6]; [11]; [30]). These activities require limited space, can be implemented with modified equipment to accommodate various developmental stages, and offer opportunities to develop coordination, agility, reaction time, and strategic thinking ([30]).

Moreover, racket sports promote cooperation and communication, supporting social and emotional learning in inclusive school environments ([48]). TT, in particular, presents unique advantages for school-based PE ([34]; [45]; [49]). Its simple rules, low cost, minimal spatial requirements, and reduced risk of injury make it highly suitable for diverse educational settings, including schools with limited facilities ([41]; [42]). Empirical evidence indicates that regular TT practice improves motor coordination, visual perception, and cognitive functions in children and adolescents ([5]). Furthermore, studies have shown benefits for children with physical disabilities, mild intellectual impairments, and neurodevelopmental conditions, making TT a promising activity for inclusive education ([4]; [18]; [26]). Longitudinal research also suggests positive effects on body composition, bone health, and overall fitness among school-aged children who regularly participate in TT programs ([1]; [40]).

Despite these benefits, the integration of TT into primary school PE curricula remains inconsistent. International reports highlight disparities in its adoption across regions, school types, and institutional contexts ([28]). While some research has examined student participation and motivation toward racket sports, less is known about teachers’ attitudes and the structural or professional factors that may facilitate or hinder their implementation ([32]). Teacher-related variables—such as sex, age, teaching experience, and administrative status—have been suggested as potential determinants of sport selection and pedagogical practices in PE, yet empirical evidence is limited ([32]). Addressing this gap is essential for informing strategies that promote equitable and evidence-based inclusion of TT in primary education.

Understanding teachers’ perceptions of TT, its educational value, and its safety profile is particularly important for designing professional development programs and allocating resources effectively. Therefore, the objective of this study was to assess the prevalence of TT implementation in primary school PE classes in Spain and to explore its association not only with teachers’ demographic and professional characteristics (sex, age, teaching experience, and administrative status) but also with their perceptions regarding injury risk, safety, inclusivity for students with physical disabilities or special educational needs according to current educational legislation in the country where the study was conducted ([36]), student engagement, and the educational value of TT in PE curricula.

## 2. Materials and Methods

### 2.1. Study Design

This research followed a quantitative, descriptive, and cross-sectional design, analyzing PE specialized teachers’ opinions regarding the use of table tennis in primary school PE lessons (6–12 years old). The sampling procedure was based on a non-probabilistic approach, relying on school accessibility. While this strategy facilitated participation and the logistics of data collection, it entails limitations in the generalization of findings to broader populations. To minimize potential selection bias, schools from all three provinces of the Autonomous Community of Aragón (Huesca, Teruel, and Zaragoza) were included. The sample encompassed institutions with diverse urban and rural settings and heterogeneous socioeconomic characteristics, incorporating public, state-subsidized, and private schools. This ensured a broad and diverse representation of the study population.

### 2.2. Participants

A total of 393 primary school PE teachers from Aragón (Spain) participated in the study. The surveyed teachers were aged between 23 and 59 years (39.6 ± 9.2 years) and had between 1 and 39 years of teaching experience in PE (13.4 ± 9.1 years). Of the total sample, 35.8% of the participating teachers were women (*n* = 176) and 44.1% were men (*n* = 217). Regarding the type of school, 49.2% of the teachers worked in public schools (*n* = 242), 28.3% in state-subsidized private schools (*n* = 139), and 2.4% in fully private schools (*n* = 12). Concerning professional roles, 46.5% of participants were teachers (*n* = 239), 20.5% tutors (*n* = 129), and 4.9% held other roles (head teachers, school secretaries…; *n* = 24). Administrative status distribution showed that 59.1% of the sample held a permanent PE post (*n* = 292), 14.8% were interim teachers (*n* = 75), 1.4% were awaiting placement (*n* = 7), and 3.9% reported other statuses (*n* = 19). Finally, the geographical location of the schools where the teachers worked was 28.7% in rural areas (*n* = 141) and 51.2% in urban areas (*n* = 252). Considering the total population of primary school PE teachers in Aragón during the study year (*n* = 1024), the final sample of 393 respondents ensured a confidence level of 95% with a margin of error of 3.89%, as calculated using the standard sample size formula for finite populations. For reference, a margin of error of 5% would have required only 280 responses. Therefore, the sample size not only meets but exceeds the minimum required for statistical validity, representing a robust dataset for the objectives of this study.

### 2.3. Inclusion and Exclusion Criteria

The selection procedure, questionnaire administration, and data collection were carried out during the 2023–2024 academic year. To ensure the proper development of the study, specific inclusion and exclusion criteria were defined. Eligible participants were physical education teachers working in primary schools (students aged 6 to 12 years) within the Autonomous Community of Aragón. Only teachers who provided informed consent were included in the final sample. Schools and teachers were excluded from the study if they did not offer physical education as part of their curriculum or if the questionnaires were incomplete or contained inconsistent responses.

### 2.4. Instruments

The instrument used was the Racket Sports Attitude Scale (RSAS) ([38]), a validated Spanish questionnaire designed to assess the practice of racket sports, particularly TT, as educational content in PE classes. The RSAS was administered exclusively in its online version, accessible via the Google Forms platform. The RSAS consists of 45 items divided into six structured subscales: General Questions (13 items), Difficulties in Applying Physical Education Content (6 items), Positive Attitudes toward Racket Sports (5 items), Benefits of TT (7 items), Facilitators for the Implementation of TT (7 items), and Barriers to the Implementation of TT (7 items). The questionnaire begins with introductory information providing instructions for proper completion, a description of the study, details of the research team, and the study’s objectives, among other relevant elements. At the end of the questionnaire, additional information is provided, where each participant, after reviewing and reading all prior content, must agree to the privacy policy regarding personal data handling and the anonymization process before participating in the research. The RSAS prompts teachers to reflect on various issues related to PE, racket sports, and more specifically, TT. The psychometric technique used to collect responses was a 4-point Likert scale without a neutral option (neither agree nor disagree), requiring respondents to select either a positive or negative stance ranging from (1) strongly disagree to (4) strongly agree (29 items). This was complemented by multiple-choice questions with multiple answers (4 items), multiple-choice questions with a single answer (10 items), and open-ended questions (2 items), where participants were voluntarily invited to provide more reflective and detailed responses if they felt it necessary for a deeper analysis of any topic of interest or relevance.

### 2.5. Procedure

The RSAS questionnaire was administered online through the Google Forms platform, exclusively in its Spanish version ([38]). Data related to the sample characteristics and participants’ responses were collected via this platform between 9 January 2024 and 28 June 2024. Through the platform, participants were required to register by providing sociodemographic information (such as sex, age, education, teaching experience, and type of educational institution, among others), although not all sociodemographic variables were necessary for the study. Once registered and after accepting the privacy policy regarding personal data handling and the anonymization process, participants completed the RSAS questionnaire. Throughout the entire research procedure, the ethical principles outlined in the Declaration of Helsinki ([47]) and the Oviedo Convention ([10]) were followed. The Ethics Committee of the Government of Aragon (ID: 12/2021) reviewed and approved the study.

### 2.6. Statistical Analysis

Descriptive statistics were calculated to summarize participant characteristics and the prevalence of TT implementation in PE classes. Categorical variables (sex, age group, and teaching experience) were expressed as frequencies and percentages. Associations between the use of TT (yes/no) and teacher characteristics (sex, age, and teaching experience) were analyzed using Pearson’s Chi-square test. For each significant association, Cramer’s V and Phi coefficient were reported as measures of association strength. Given the ordinal nature of age and teaching experience, Somers’ D was also calculated to assess directional associations. Additional Chi-square analyses were conducted to explore associations between TT implementation and specific teacher perceptions, including: whether TT is perceived as having a lower injury risk compared to other racket sports, its suitability for students with physical problems or special educational needs, its overall safety for primary school students, its effectiveness in fostering student interest and engagement in PE classes, and its educational value as PE content. For these analyses, measures of association (Cramer’s V, Phi, and Somers’ D for ordinal variables) were also reported to quantify effect sizes. Effect sizes were interpreted according to conventional thresholds: for Cramer’s V, values of 0.10, 0.30, and 0.50 indicated small, moderate, and large associations, respectively. For Somers’ D, values below 0.20 were considered weak, between 0.20 and 0.50 moderate, and above 0.50 strong. All statistical analyses were performed using the statistical software IBM SPSS Statistics (IBM Corp., Armonk, NY, USA), version 30.0 for the Windows operating system. Statistical significance was set at *p* < 0.05 (two-tailed). Missing data were handled through pairwise deletion to maximize the use of available information. Results are presented as contingency tables and summarized in a comparative table displaying Chi-square values, degrees of freedom, *p*-values, and effect size indices (Cramer’s V, Somers’ D).

## 3. Results

### 3.1. Descriptive Analysis

A total of 393 PE teachers participated in the study. Table 1 summarizes the demographic and professional characteristics of the sample. Most participants were male (44.1%) and worked primarily in urban environments (51.2%). Most respondents were employed in public schools (49.2%), followed by subsidized (28.3%) and private institutions (2.4%). Regarding professional roles, teachers represented 46.5% of the sample, while tutors accounted for 20.5%. In terms of administrative status, 59.1% held permanent positions, 14.8% were interim, and 1.4% were awaiting placement.

The mean age of respondents was 39.6 years (SD = 9.16), and their average teaching experience was 13.46 years (SD = 9.11) (Table 2).

Among the 393 teachers surveyed, only 9.3% (*n* = 46) reported incorporating TT into their PE lessons, whereas 70.5% (*n* = 347) stated they did not include it. Statistical analyses were performed to explore associations between the adoption of TT and teacher characteristics (sex, age, and teaching experience).

### 3.2. Sex Differences

The prevalence of TT use (Figure 1) was similar between men (11.8%) and women (11.6%), indicating nearly identical adoption rates across sexes. However, when considering only teachers who reported using TT, the majority were male (56.5%) compared to female (43.5%). The Chi-square test confirmed a statistically significant association between sex and TT use (χ^2^ = 492.0, df = 4, *p* < 0.001), with Cramer’s V indicating a moderate effect size (V = 0.707, *p* < 0.001).

### 3.3. Age Distribution

TT use varied significantly across age groups (χ^2^ = 27.2, df = 7, *p* < 0.001) (Figure 2). Teachers aged 41–45 years had the highest adoption rate (27.8%, accounting for 43.5% of all users), followed by those aged 36–40 years (13.0%) and 56–60 years (13.8%). Younger teachers (20–25 years) reported no use, and adoption remained low (<10%) in teachers under 35. The association was positive but weak (Somers’ D = 0.071, *p* = 0.021; Cramer’s V = 0.263, *p* < 0.001).

### 3.4. Teaching Experience

A significant relationship emerged between teaching experience and TT use (χ^2^ = 30.0, df = 6, *p* < 0.001) (Figure 3). Teachers with 16–20 years of experience showed the highest adoption (31.7%, representing 41.3% of all users). Intermediate experience levels (11–15 and 21–30 years) reported moderate use (10–12%), whereas early-career teachers (<10 years) had lower adoption rates (≈5%). Somers’ D confirmed a weak positive association (D = 0.099, *p* = 0.003; Cramer’s V = 0.276, *p* < 0.001).

### 3.5. Perception of Injury Risk

The perception that TT entails a lower risk of injuries compared to other racket sports (Figure 4) was significantly associated with its implementation in PE lessons (χ^2^ = 492.4, df = 8, *p* < 0.001). Among teachers using TT, 60.9% disagreed with the statement, while 13.0% strongly agreed. Non-users showed a similar pattern (56.8% disagreed, 13.3% strongly agreed). Effect size analysis indicated a strong association (Cramer’s V = 0.707, *p* < 0.001; Somers’ D = 0.547, *p* < 0.001).

### 3.6. Perceived Suitability for Students with Physical Disabilities

A significant association was found between TT use and the belief that it is a suitable sport for students with physical impairments (χ^2^ = 499.3, df = 8, *p* < 0.001) (Figure 5). Among users, 60.9% disagreed with the statement, whereas 23.9% strongly agreed. Non-users exhibited similar perceptions (65.1% disagreed, 14.7% strongly disagreed). The association was strong (Cramer’s V = 0.712, *p* < 0.001; Somers’ D = 0.594, *p* < 0.001).

### 3.7. Perceived Suitability for Students with Special Educational Needs

Teachers’ perception of TT (Figure 6) as a recommended activity for students with special educational needs was also significantly associated with its implementation (χ^2^ = 502.9, df = 8, *p* < 0.001). Among those who used table tennis, 65.2% disagreed, 23.9% strongly agreed, and none strongly disagreed. Non-users similarly reported disagreement (64.8%) or strong disagreement (6.3%). Effect size measures indicated a strong association (Cramer’s V = 0.715, *p* < 0.001; Somers’ D = 0.578, *p* < 0.001).

### 3.8. Perceived Safety in Primary Education

The percentage values obtained for the variable perceived safety by teaching staff in the use of TT as educational content are presented in Figure 7. In both cases (users and non-users) there is a high level of responses in favor of considering TT as a very safe sport to be used in PE classes in primary education with values above 56%.

### 3.9. Perceived Student Engagement

The belief that TT facilitates student interest and engagement in PE was significantly associated with its use (χ^2^ = 499.3, df = 8, *p* < 0.001) (Figure 8). Among teachers using TT, 60.9% disagreed, 23.9% strongly agreed, while only 4.3% strongly disagreed. Non-users had similar patterns (65.1% disagreed, 6.9% strongly disagreed). The association was strong (Cramer’s V = 0.712, *p* < 0.001; Somers’ D = 0.594, *p* < 0.001).

### 3.10. Perception of TT as a Recommended PE Content

Finally, the perception of tennis as a highly recommended content for primary school PE was also significantly linked to TT implementation (χ^2^ = 504.9, df = 8, *p* < 0.001) (Figure 9). Among users, 37.0% agreed, 17.4% strongly agreed, and 43.5% disagreed. Non-users were mostly in agreement (55.3%) or strongly agreed (13.8%), while 23.9% disagreed. This association was strong (Cramer’s V = 0.716, *p* < 0.001; Somers’ D = 0.586, *p* < 0.001).

Overall, the analyses indicate that TT is infrequently implemented in PE curricula. Its adoption is slightly more common among teachers in middle adulthood with moderate teaching experience. Significant associations were also found with perceptions of safety, inclusivity, and student engagement, suggesting that these factors are statistically linked to the decision to include TT in PE lessons. A summary of these results is presented in Table 3.

## 4. Discussion

The objective of this study was to assess the prevalence of TT implementation in primary school PE classes in Spain and to explore its association with teachers’ perceptions of TT as an educational and health-promoting content within the PE curriculum. Specifically, the study examined how the perceptions related to injury risk, safety, inclusivity for students with physical disabilities or special educational needs, student engagement, and overall educational value are associated with TT use. In addition, the analysis considered key demographic and professional characteristics of teachers, including gender, age, teaching experience, and administrative status. This study provides novel insights into the current integration of TT within primary school PE curricula and the teacher-related factors influencing its adoption. Despite its recognized benefits, including low injury risk, inclusivity, and suitability for limited spaces, TT remains underrepresented, with fewer than 12% of surveyed teachers incorporating it into their lessons. This finding aligns with previous reports indicating that racket sports, particularly TT, are underutilized in educational settings compared to more traditional activities such as football, athletics, or basketball ([7]; [44]; [49]). Our results highlight significant associations between the implementation of TT and teachers’ age and teaching experience.

Middle-aged teachers (41–45 years) and those with 16–20 years of teaching experience were more likely to include TT in their PE classes. This pattern may reflect accumulated pedagogical expertise, greater familiarity with diverse teaching methods, and increased confidence in adapting PE content to different school environments. Early-career teachers (<10 years of experience) and younger respondents (<35 years old) reported the lowest adoption rates, suggesting potential gaps in initial teacher training or limited exposure to racket sports during their own education, although there may be other reasons for this lack of implementation (e.g., an annual teaching programme in which they have not been previously included, supervision by a mentor teacher or the perception that it may be considered a risky sport to incorporate at the primary school stage), the lack of implementation may be due to the fact that the sport is not included in the annual teaching programme. These findings emphasize the need for targeted professional development programs to increase competence and confidence in teaching TT and other racket sports ([2]; [19]).

From a behavioral perspective, these differences may also be interpreted within established models of teacher behavior and motivation. According to the Theory of Planned Behavior and Self-Determination Theory, teachers’ decisions to implement innovative or less familiar PE activities depend on perceived behavioral control (confidence and resources), subjective norms (colleagues’ or institutional expectations), and intrinsic motivation to promote health-oriented education. Experienced teachers may feel more autonomous and competent to integrate TT, while newly qualified teachers, often more constrained by curricular pressures or limited material resources, may exhibit lower behavioral intention to innovate. Understanding these motivational mechanisms could inform targeted behavioral interventions aimed at encouraging TT adoption through increased support, autonomy, and professional recognition.

Sex differences were not statistically meaningful in practical terms, as both male and female teachers exhibited similarly low implementation rates. This suggests that barriers to adopting TT are systemic rather than gender-specific, possibly linked to structural limitations (e.g., lack of facilities, equipment shortages) or curricular priorities that favor other sports ([22]; [24]). Importantly, despite low adoption rates, teachers overwhelmingly perceived TT as a safe, educationally valuable activity suitable for students with physical or special educational needs. The strong effect sizes observed between TT adoption and positive teacher perceptions (Cramer’s V and Somers’ D > 0.70) underscore this attitudinal support. This discrepancy between perception and practice mirrors previous findings on inclusive sports and health-promoting activities in school PE ([6]; [43]).

The perceptions expressed by teachers through the RSAS regarding health-related aspects in school-age children clearly indicate a shared view that TT is highly beneficial for use in PE classes, primarily due to its low injury risk and high safety during practice ([29]). These findings align with the existing literature, which indicates that racket sports are associated with fewer injuries compared to many other disciplines, with a reported injury incidence as low as 0.01, or one injury per 1000 h of practice ([30]). Nonetheless, there is a notable lack of research specifically addressing the injury incidents of TT. Some studies suggest that injuries in TT are infrequent, with occurrence rates increasing at the competitive level as practice volume rises ([29]), while remaining relatively rare in educational settings. For this reason, TT is considered a valuable sport for promoting healthy lifestyles from an early age ([22]).

Regarding the inclusion of students with physical disabilities or special educational needs in PE classes, teachers widely perceive TT as an accessible, inclusive, and educationally appropriate activity. In this respect, recent research supports the use of adapted TT as an effective strategy to increase physical activity levels and promote safe, motivating, and accessible physical exercise among students with disabilities ([26]). Additionally, its low equipment cost, adaptability to diverse functional abilities, and potential for developing motor and coordination skills make TT an ideal content for comprehensive and inclusive PE, fostering both interest and active participation among students ([3]; [4]; [27]; [39]; [45]).

From a pedagogical perspective, integrating TT into PE curricula may enhance children’s motor coordination, reaction time, and cognitive engagement, while also promoting social interaction and cooperation ([14]; [15]; [26]). Its minimal space and equipment requirements make it particularly suitable for schools with limited resources ([41]; [42]). Given these attributes and the overall positive perceptions among teachers, promoting TT adoption may require systemic support through (i) curriculum guidelines that explicitly recommend racket sports; (ii) increased access to low-cost equipment and adaptable facilities; and (iii) targeted in-service training focused on practical strategies for integrating TT into PE contexts ([20], [21]).

The study’s strengths include its large sample size and the use of a validated instrument, RSAS, allowing for a robust examination of demographic and professional factors. Another strength lies in its originality: this research applies a validated scale to a context rarely explored, table tennis in primary school physical education, thus filling a gap in the literature on teachers’ behavioral and attitudinal factors in sport-specific pedagogy. The mixed methodological, applied approach enhances the credibility of the findings and provides a replicable model for future investigations in other educational contexts. However, certain limitations must be acknowledged. First, the cross-sectional design precludes causal inference regarding the factors influencing adoption. Second, self-reported data may be subject to recall or social desirability biases. Finally, the study focused exclusively on primary school teachers in a single area in the north of Spain (Aragón), limiting the generalizability of findings to other educational contexts or age groups. Future research should extend this line of inquiry by examining longitudinal trends in racket sport implementation and exploring behavioral frameworks that explain why teacher motivation, confidence, and professional experience influence the inclusion of sports such as table tennis. Studies might also assess how targeted teacher training, resource allocation, or policy-level interventions can enhance adoption. Additionally, intervention research could evaluate how integrating table tennis into PE impacts children’s physical literacy, motor skills, engagement, and lifelong physical activity behaviors ([5]; [17]; [25]; [37]). From a broader perspective, the contribution of this work extends beyond the empirical findings. Methodologically, it demonstrates the value of structured assessment tools such as the RSAS for analyzing teacher behavior. Practically, it provides evidence-based insights that can inform curriculum design, promote inclusive and health-oriented physical education, and guide professional development strategies in both national and international settings.

## 5. Conclusions

In conclusion, this study highlights both the growing interest in TT within primary school PE and the persistent gap between its perceived educational value and its practical implementation. While teachers continue to view the sport as safe, inclusive, and pedagogically valuable, its integration into PE curricula remains inconsistent. Notably, the results indicate that younger teachers are increasingly inclined to include TT, potentially signaling a generational shift in pedagogical approaches. However, structural barriers—such as limited access to equipment or insufficient institutional support—still hinder widespread adoption. These findings underscore the need for targeted investments in teacher training, resource allocation, and curricular innovation to align educational practices with the inclusive and health-promoting potential of racket sports like TT. They also provide actionable insights for policy makers, school leaders, and teacher education programs committed to diversifying and modernizing physical education.

## Figures and Tables

**Figure 1 behavsci-15-01495-f001:**
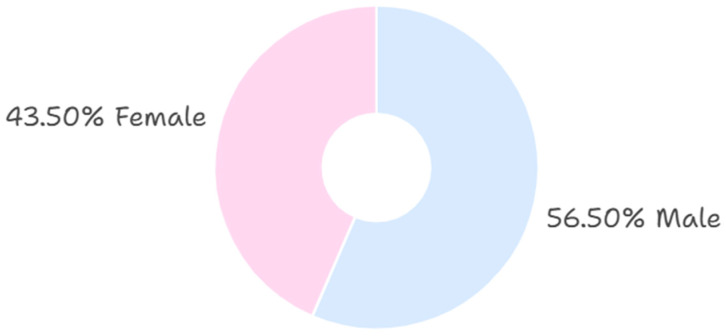
Distribution of TT users by gender.

**Figure 2 behavsci-15-01495-f002:**
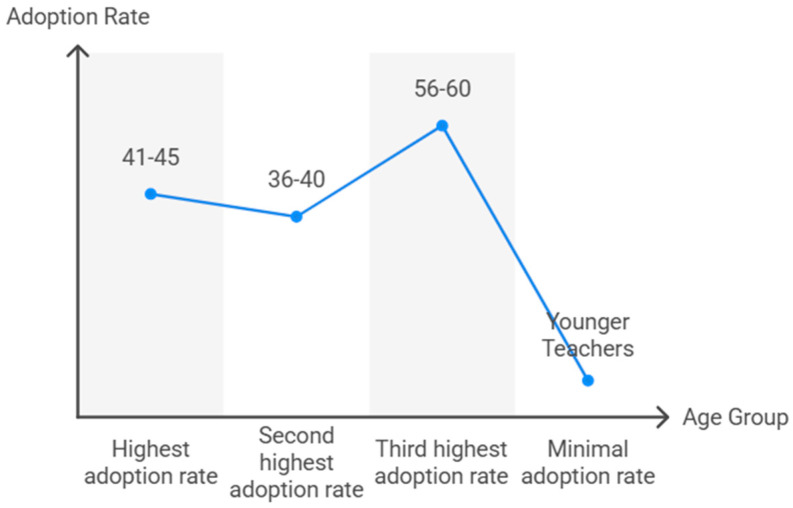
TT adoption rates by age group.

**Figure 3 behavsci-15-01495-f003:**
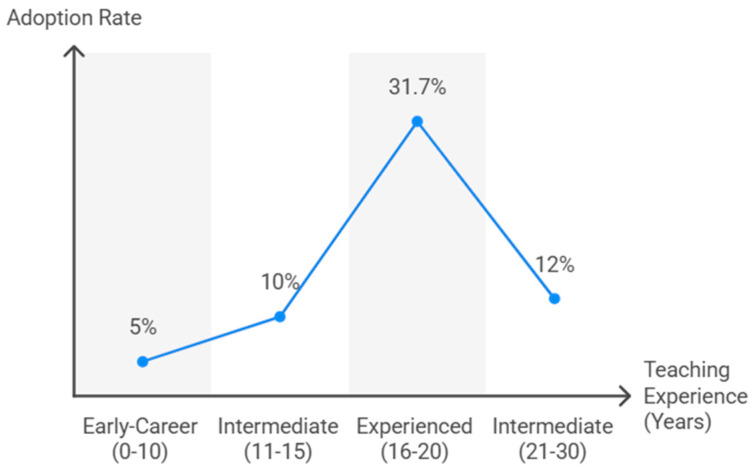
TT adoption rates by teaching experiences.

**Figure 4 behavsci-15-01495-f004:**
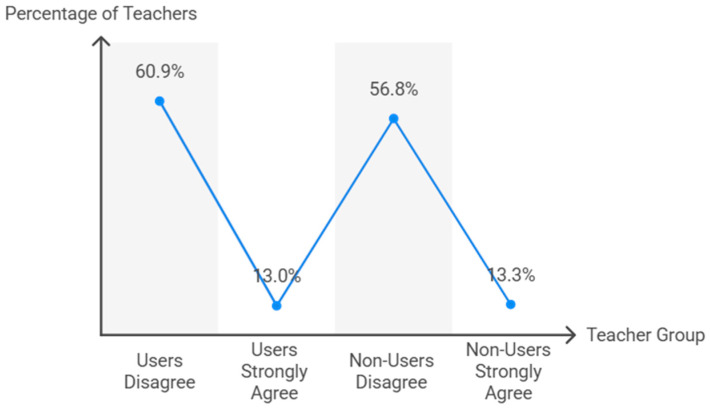
Teachers’ perception of table tennis injury risk.

**Figure 5 behavsci-15-01495-f005:**
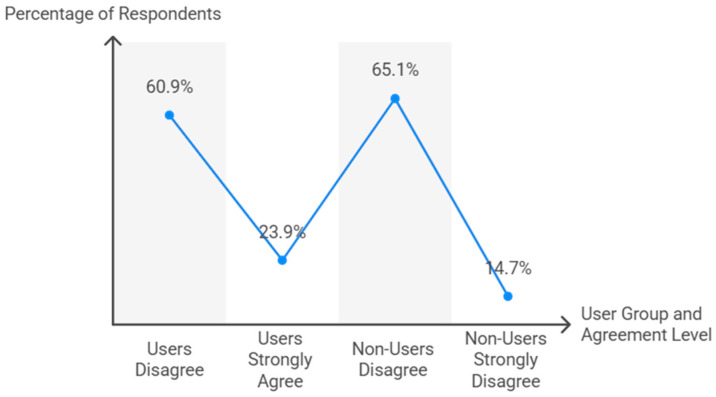
Perceptions of TT suitability for students with physical impairments.

**Figure 6 behavsci-15-01495-f006:**
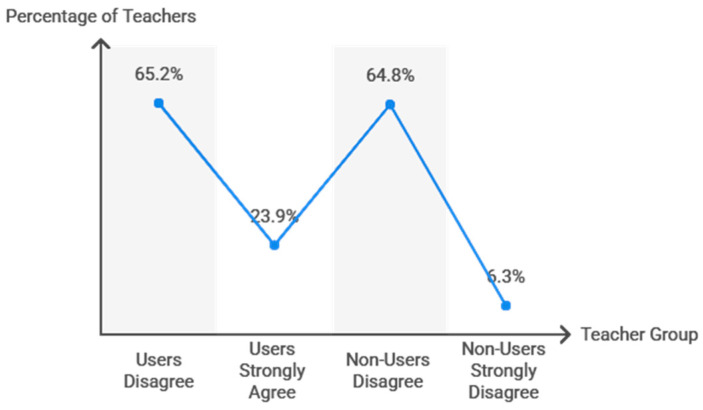
Teachers’ perception of table tennis for SEN students.

**Figure 7 behavsci-15-01495-f007:**
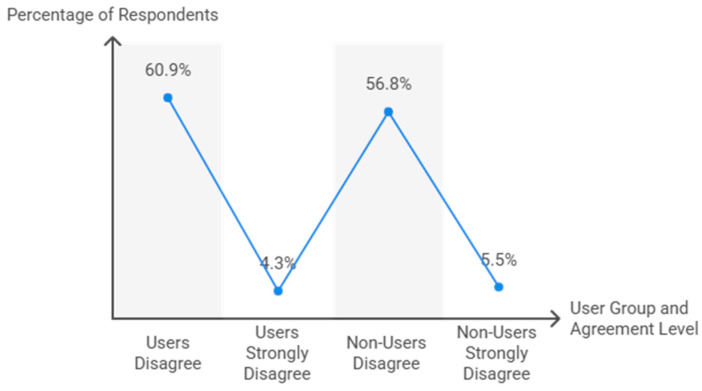
Perceived safety of table tennis among primary school students.

**Figure 8 behavsci-15-01495-f008:**
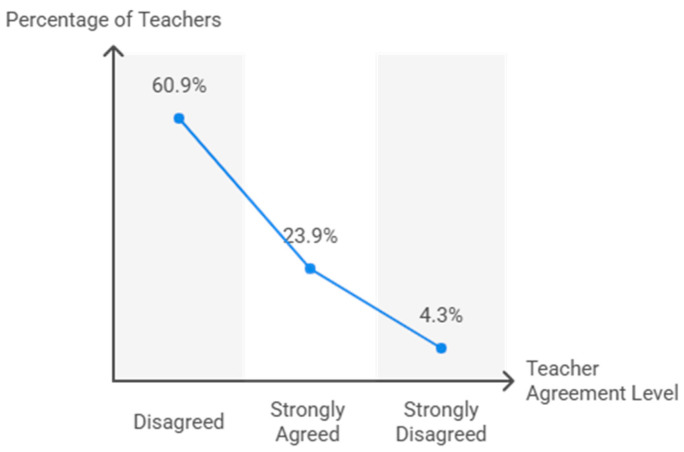
Teachers’ perceptions of TT engagement.

**Figure 9 behavsci-15-01495-f009:**
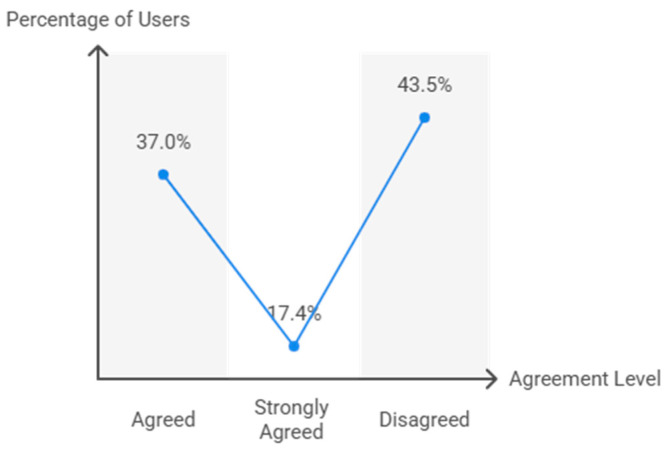
Perception of TT as recommended PE content.

**Table 1 behavsci-15-01495-t001:** Descriptive statistics of the sample.

Variable	Category	Frequency (*n*)	Percentage (%)
Sex	Female	176	35.8
	Male	217	44.1
School environment	Urban	252	51.2
	Rural	141	28.7
School type	Public	242	49.2
	Private	12	2.4
	Subsidized	139	28.3
Professional role	Teacher	239	46.5
	Tutor	129	20.5
	Other	24	4.9
Administrative status	Awaiting placement	7	1.4
	Interim	75	14.8
	Permanent Physical Education post	292	59.1
	Other	19	3.9

**Table 2 behavsci-15-01495-t002:** Cumulative descriptive statistics (age and teaching experience) of the sample.

Variable	Mean	SD
Age (years)	39.6	9.16
Teaching experience (years)	13.4	9.11

**Table 3 behavsci-15-01495-t003:** Cumulative summary of results.

Variable	χ^2^ (df)	*p*-Value	Association Strength
Sex	492.0 (4)	<0.001	Cramer’ V = 0.707
Age group	27.2 (7)	<0.001	Somers’ D = 0.071
Teaching experience	30.0 (6)	<0.001	Somers’ D = 0.099
Perception of Injury Risk	496.1 (8)	<0.001	Somers’ D = 0.575
Perceived Suitability for Students with Physical Disabilities	505.6 (8)	<0.001	Somers D = 0.599
Perceived Suitability for Students with Special Educational Needs	502.9 (8)	<0.001	Somers’ D = 0.578
Perceived Safety in Primary Education	492.4 (8)	<0.001	Somers’ D = 0.574
Perceived Student Engagement	499.2 (8)	<0.001	Somers’ D = 0.594
Perception of Table Tennis as a Recommended PE Content	504.9 (8)	<0.001	Somers’ D = 0.586

## Data Availability

All data can be obtained from the corresponding author upon reasonable request.

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
