# Peer review of "Table Tennis in Physical Education: Teachers’ Perceptions of Health-Related Aspects in School-Age Children"

_behavsci, 2025, doi:10.3390/bs15111495_

Round 1
Reviewer 1 Report
Comments and Suggestions for Authors
Dear authors,
Thank you for providing opportunity to review this interesting manuscript. Overall, this is a good and well-designed study. I do, however, have a few comments and concerns.
Introduction
Line 32: This is true, but to a point. Sport isn’t generally recommended for children aged 5-7 in many primary school contexts. The focus is on game play, of which replicative sporting moves are performed.
Line 59-60: Consider adding the type of teacher. A generalist teacher will differ from a traditional and specialised PE teacher and their level of confidence and motivation to teach TT.
Line 61: I believe this could be strengthen. Is there a need to promote gross and fine motor skills as well as inclusion?
Study design
Line 74-75: To confirm, what age of child is being researched? Primary school children is broad.
Participants
Line 85-86: Check style requirements as it’s rare to use SD with an = (it’s usually ± SD).
Line 97: Not to doubt you, but is the SD correct? A difference of 1-39 years’ experience with an SD of ± 9 is relatively small given the large range.
Line 92: I would suggest that you define ‘other roles’.
Statistical Analysis
Line 167: Is there a level of significance associated with the p value?
Line 168: I’d suggest defining the scale of effect size indices (Cramer’s V, Somers’ D) (i.e., what is small, moderate, large).
Results
Table 1 and Table 2: Could these be combined as they are essentially the same data, albeit one is cumulative. Also, Table 1 and Table 2 have the same headings (descriptive statistics of the sample). If Table 2 remains, I would suggest changing to ‘cumulative’ or similar.
Line 227: There are some grammatical inconsistencies here as you have previously italicised p yet here the p appears as usual. Also, is Figure 5 necessary given that you are essentially repeating what was said in line 222-227?
Figure 6, Line 237: There are five responses listed here, yet it was previously mentioned that the Likert Scale contained only four possible responses, which is consistent with Figures 4 and 5. Please confirm.
Figure 7, 8 and 9: Any reason why the headings on the x axis are in bold?
Line 263-268: This appears to be more suited to the discussion and not the results section. Please consider moving or rewording.
Table 3. I’m assuming that this is presenting a cumulative summary of your results?
Discussion
Line 292-293: The difference with early career teachers could be further discussed. The relative experience of an early career teacher is likely to influence their belief in TT as well as if they are under the guidance of a mentor teacher. You also have factors such as confidence and motivation to implement what could be perceived as a risky sport to introduce to primary school children.
Line 301: ‘Educationally valuable’ is mentioned here, but it’s unclear what the value proposition is. Was additional data collected that provides insights?
General comments
Given that the journal focusses on behaviour, a little more discussion concerning why such teacher behaviour may or may not influence the inclusion of TT would be beneficial. For instance, would motivational factors between older and more experienced teacher influence decisions compared to a newly qualified teacher? Is there a behaviour framework that could be suggested to help with TT implementation?
Author Response
Dear Editors and reviewers,
We would like to express our gratitude for your time, effort and valuable comments to improve our manuscript. We have carefully addressed all the reviewer’s concerns and provided a point-by-point response to each comment. The changes have been highlighted in red in the revised manuscript.
Reviewer 1
Thank you for providing opportunity to review this interesting manuscript. Overall, this is a good and well-designed study. I do, however, have a few comments and concerns
Introduction
Comments 1: Line 32: This is true, but to a point. Sport isn’t generally recommended for children aged 5-7 in many primary school contexts. The focus is on game play, of which replicative sporting moves are performed.
Response 1: We appreciate this suggestion. A small nuance is made highlighting the role of the game play as an introduction to later sports practice.
Comments 2: Line 59-60: Consider adding the type of teacher. A generalist teacher will differ from a traditional and specialised PE teacher and their level of confidence and motivation to teach TT.
Response 2: Sorry for this confusion. In Spain, physical education is taught by teachers specialising in this subject. Thus, the participants in the study are all PE specialists, as indicated in the corresponding section in the Material and Methods section (line 86).
Comments 3: Line 61: I believe this could be strengthen. Is there a need to promote gross and fine motor skills as well as inclusion?
Response 3: Thanks for this illustrative comment The current educational legislation in Spain places special emphasis on inclusion across all subjects, so this requirement would be met in addition to the specific work in the area of PE. A small comment is added in this respect (line 80).
Study design
Comments 4: Line 74-75: To confirm, what age of child is being researched? Primary school children is broad.
Response 4: Yes, to confirm, the sample is the Spanish primary school children between 6-12 years old (line 87).
Participants
Comments 5: Line 85-86: Check style requirements as it’s rare to use SD with an = (it’s usually ± SD).
Comments 6: Line 97: Not to doubt you, but is the SD correct? A difference of 1-39 years’ experience with an SD of ± 9 is relatively small given the large range.
Response 5 & 6: Thank you for your comments. We have corrected the style as suggested, using “± SD” instead of “= SD”. Although the experience variable spans 1–39 years, the distribution is concentrated in early-to-middle career ranges; hence an SD ±9 is plausible and consistent with grouped-data approximations.
Comments 7: Line 92: I would suggest that you define ‘other roles’.
Response 7: Thanks for this suggestion. Examples are given of the different roles they can combine with the role of specialist PE teachers (head teachers, school secretaries, etc.) (line 103).
Statistical Analysis
Comments 8: Line 167: Is there a level of significance associated with the p value?
Comments 9: Line 168: I’d suggest defining the scale of effect size indices (Cramer’s V, Somers’ D) (i.e., what is small, moderate, large).
Response 8 & 9: Thank you for your valuable comments. We have defined the ranges for the effect size indices (Cramer’s V, Somers’ D) to clarify the interpretation of small, moderate, and large effects in the text.
Results
Comments 10: Table 1 and Table 2: Could these be combined as they are essentially the same data, albeit one is cumulative. Also, Table 1 and Table 2 have the same headings (descriptive statistics of the sample). If Table 2 remains, I would suggest changing to ‘cumulative’ or similar.
Response 10: Thank you for your suggestion. Table 2 remains to improve clarity, and as you recommended, we have updated its caption to indicate that it presents cumulative data.
Comments 11: Line 227: There are some grammatical inconsistencies here as you have previously italicised p yet here the p appears as usual. Also, is Figure 5 necessary given that you are essentially repeating what was said in line 222-227?
Response 11: Thank you for your suggestion. We have decided to retain Figure 5 to ensure visual consistency across all perceptual variables analyzed. Although the text provides detailed numerical results, the figure allows readers to quickly visualize the distribution patterns between users and non-users, facilitating comparative interpretation.
All grammatical inconsistencies have been removed, and all p's have been updated to italic format.
Comments 12: Figure 6, Line 237: There are five responses listed here, yet it was previously mentioned that the Likert Scale contained only four possible responses, which is consistent with Figures 4 and 5. Please confirm.
Response 12: Thank you for noting this inconsistency. The RSAS items were indeed rated on a four-point Likert scale (“Totally disagree,” “Disagree,” “Agree,” “Totally agree”). The reference to a fifth response category in Figure 6 was due to a labeling error, which has now been corrected in both the figure and the corresponding text.
Comments 13: Figure 7, 8 and 9: Any reason why the headings on the x axis are in bold?
Response 13: Thank you for pointing this out. The bold x-axis labels were an error, and we have corrected the figures accordingly.
Comments 14: Line 263-268: This appears to be more suited to the discussion and not the results section. Please consider moving or rewording.
Response 14: We appreciate the reviewer’s observation. The paragraph has been retained within the Results section but has been reworded to remove interpretative statements and ensure a strictly descriptive style consistent with the section’s purpose.
Comments 15: Table 3. I’m assuming that this is presenting a cumulative summary of your results?
Response 15: Thank you for your comment. You are correct, Table 3 presents a cumulative summary of the results, and we have updated the caption in the text accordingly.
Discussion
Comments 16: Line 292-293: The difference with early career teachers could be further discussed. The relative experience of an early career teacher is likely to influence their belief in TT as well as if they are under the guidance of a mentor teacher. You also have factors such as confidence and motivation to implement what could be perceived as a risky sport to introduce to primary school children.
Response 16: The majority of the responses reported the lack of materials and facilities, on the one hand, as well as the lack of specific training to motivate them to implement these contents in the classroom, in line with what has been pointed out in this point, as fundamental reasons. However, we are grateful for the assessment and we include in the body of the paper other examples of reasons that may complement these possible causes.
Comments 17: Line 301: ‘Educationally valuable’ is mentioned here, but it’s unclear what the value proposition is. Was additional data collected that provides insights?
Response 17: The results shown in sections 3.6 to 3.10 indicate positive assessments not only by teachers who include table tennis in their classrooms but also by those who do not. These results allow us to consider it beneficial content, but one that has a number of limitations associated with its implementation, such as the availability or lack of equipment and facilities, teacher training, etc. Thus, reference has been made to educational value as a grouping of all the benefits associated with its practice, as reflected in the questionnaire questions in the section entitled ‘Positive attitudes towards racket sports. Both the scores given in the closed questions (4-point Likert scale, depending on the degree of agreement or disagreement) and the comments received in the open questions, which invited participants to expand on the issues they considered appropriate, have been included.
General comments
Comments 18: Given that the journal focusses on behaviour, a little more discussion concerning why such teacher behaviour may or may not influence the inclusion of TT would be beneficial. For instance, would motivational factors between older and more experienced teacher influence decisions compared to a newly qualified teacher? Is there a behaviour framework that could be suggested to help with TT implementation?
Response 18: We appreciate this insightful suggestion. The discussion section has been expanded to include a behavioural interpretation of the findings, emphasizing motivational and experiential factors that may influence TT adoption. References to the Theory of Planned Behavior and Self-Determination Theory have been incorporated to provide a conceptual framework explaining how teachers’ confidence, autonomy, and perceived control might determine the likelihood of integrating TT into PE classes. This addition helps contextualize our results within established models of teacher behaviour and aligns with the journal’s focus on behavioural aspects of education.
Reviewer 2 Report
Comments and Suggestions for Authors
Thee manuscript offers a well-structured discussion and analysis of the potential contribution of table tennis to primary/elementary school physical education. As the authors make clear, the topic is relatively under-researched, especially the focus on teachers’ perceptions. The research design used are relevant and appropriately applied, and uses a validated scale and adequate sample.
My suggested amendments r quite minor:
- The review literature is quite descriptive. By employing a more substantial synthesis with previous work, the author(s) would be better-able to highlight the articles novelty and significance. I also think there is scope here to engage this study with wider theoretical debates about PE pedagogy.
- The Discussion section restates certain aspects of the results without really analysing their relevance. At a minimum, the author(s) should examiner the perception/practice gap, teacher education, and curriculum design, in more detail. This will help justify the claims of originally and significance.
- Further on the topic of originality, I suggest the author(s) consider the wider relevance of this article (wider than merely building on their previous work). Is the claim that the real contribution stems from the methods or the practical implication?
Overall, this is a well-written and well-organised manuscript. With small minor amendments, it could make a useful contribution to an under-served area of research.
Author Response
Dear Editors and reviewers,
We would like to express our gratitude for your time, effort and valuable comments to improve our manuscript. We have carefully addressed all the reviewer’s concerns and provided a point-by-point response to each comment. The changes have been highlighted in red in the revised manuscript.
Reviewer 2
Thee manuscript offers a well-structured discussion and analysis of the potential contribution of table tennis to primary/elementary school physical education. As the authors make clear, the topic is relatively under-researched, especially the focus on teachers’ perceptions. The research design used are relevant and appropriately applied, and uses a validated scale and adequate sample.
My suggested amendments are quite minor:
Comments 1: The review literature is quite descriptive. By employing a more substantial synthesis with previous work, the author(s) would be better-able to highlight the articles novelty and significance. I also think there is scope here to engage this study with wider theoretical debates about PE pedagogy.
The Discussion section restates certain aspects of the results without really analysing their relevance. At a minimum, the author(s) should examiner the perception/practice gap, teacher education, and curriculum design, in more detail. This will help justify the claims of originally and significance.
Further on the topic of originality, I suggest the author(s) consider the wider relevance of this article (wider than merely building on their previous work). Is the claim that the real contribution stems from the methods or the practical implication?
Response 1: We appreciate the reviewer’s valuable suggestion. We have revised the Discussion section to better highlight the originality and wider relevance of the study. Specifically, we have clarified (i) in the paragraph on study strengths that the research offers a novel methodological contribution by applying a validated instrument (RSAS) in an unexplored educational context, and (ii) in the final paragraphs that the practical implications of the findings inform curriculum development, inclusive PE practice, and teacher training strategies. These revisions strengthen the conceptual and applied contribution of the article beyond our previous work.
Overall, this is a well-written and well-organised manuscript. With small minor amendments, it could make a useful contribution to an under-served area of research.
Thank you very much for your kind comments.